# Identification of Cancerous Skin Lesions Using Vibrational Optical Coherence Tomography (VOCT): Use of VOCT in Conjunction with Machine Learning to Diagnose Skin Cancer Remotely Using Telemedicine

**DOI:** 10.3390/cancers15010156

**Published:** 2022-12-27

**Authors:** Frederick H. Silver, Arielle Mesica, Michael Gonzalez-Mercedes, Tanmay Deshmukh

**Affiliations:** 1Department of Pathology and Laboratory Medicine, Rutgers Robert Wood Johnson Medical School, The State University of New Jersey, Piscataway, NJ 08854, USA; 2OptoVibronex, LLC, Bethlehem, PA 18015, USA

**Keywords:** skin, basal cell carcinoma, squamous cell carcinoma, melanoma, fibrosis, machine learning, cancer, artificial intelligence, vibrational OCT, resonant frequency, stiffness, elastic modulus

## Abstract

**Simple Summary:**

Skin cancer detection is an important problem since it is the most common form of cancer in the US and the number of cases is increasing in the US and worldwide. Early detection of these cancers can limit their ability to: (1) disseminate throughout the body, (2) cause illnesses or (3) even cause premature death. We have used audible sound from a speaker at different frequencies to vibrate the skin and then used low intensity red light to determine how the different sound waves displaced the skin. The amount of displacement is related to the stiffness of each tissue component. Cancerous tissues are found to be stiffer than normal skin and the degree of stiffness can be used to differentiate between normal skin and skin cancers. The results of these studies indicate that skin cancers can be detected remotely using a device to vibrate and measure the displacement of tissues at different frequencies. Use of computer techniques and remote testing can facilitate identification of skin cancers in areas underserved by Dermatologists. Since this technology can detect skin cancers as small as 0.1 mm early detection will limit the undesirable effects of skin cancer.

**Abstract:**

In this pilot study, we used vibrational optical tomography (VOCT), along with machine learning, to evaluate the specificity and sensitivity of using light and audible sound to differentiate between normal skin and skin cancers. The results reported indicate that the use of machine learning, and the height and location of the VOCT mechanovibrational peaks, have potential for being used to noninvasively differentiate between normal skin and different cancerous lesions. VOCT data, along with machine learning, is shown to predict the differences between normal skin and different skin cancers with a sensitivity and specificity at rates between 78 and 90%. The sensitivity and specificity will be improved using a larger database and by using other AI techniques. Ultimately, VOCT data, visual inspection, and dermoscopy, in conjunction with machine learning, will be useful in telemedicine to noninvasively identify potentially malignant skin cancers in remote areas of the country where dermatologists are not readily available.

## 1. Introduction

Currently there are about 10 M skin biopsies conducted each year in the US, of which 44.5% are cancerous [1,2]. By 2050, it is predicted that 25 M people will be diagnosed with skin cancer in the US, suggesting that, at the current rate, about 50 M biopsies will be examined each year [1,2]. As new topical treatments are developed and techniques to noninvasively follow lesion outcomes are introduced, the rate of excisional therapy will decrease, especially for basal cell carcinomas that are localized. This will provide the patient with alternative treatment options, especially for lesions on the face, neck, and head where excisional biopsies may require subsequent plastic surgery. As the use of telemedicine is expanding, other noninvasive methods will be needed to diagnose skin cancers, especially melanomas, since the numbers of trained dermatologists are not expected to increase as rapidly as the number of skin cancers in the next 30 years.

A literature review covering the period between 1990 and 2009 reported that, while the overall rate of the management of skin disorders was equivalent using telemedicine as opposed to direct clinical contact, it was reported that teledermatology was inferior to direct patient examination for malignant lesions [3]. Currently, there are two types of teledermatological approaches. The first one involves data storage and forwarding the information, such as a photograph or quantitative information, to a clinician who reviews the data hours or days later [4]. The second approach is direct physician interactions with the patient over the internet. Additional studies from both nursing homes and other locations that are in areas where dermatologists cannot schedule to see patients in a reasonable amount of time are needed. Teledermatology is viewed positively when it is used as part of routine clinical examinations [5]. The introduction of virtual methods is expected to reduce costs and facilitate clinical trials. The centralization of data collection and a decrease in the resources needed to manage participating sites can be achieved virtually [6]. Visual inspection and dermoscopy are used to screen many of the lesions observed at the clinic [7,8,9], however, other methods are needed to identify the margins and depths of skin cancers for teledermatology to be optimized. In addition, the ability to identify small lesions will dramatically reduce the invasive potential of new cancers.

Currently, visual observation and dermoscopy are the gold standards for the diagnosis of cancerous skin lesions [9,10,11,12,13]. The dermoscopic diagnosis of BCC is based on several features, including the size and shape of blood vessels, the size, type and description of pigmented structures, and the presence of ulceration [9]. Dermoscopy improves the diagnostic sensitivity and the specificity of lesion detection compared with naked eye examination alone. However, the dermoscopic diagnosis of BCC is complicated by arborizing vessels, as well as branching, that are not present in normal skin [9]. Short fine vessels are also a characteristic of BCC.

Using dermoscopy, the diagnostic accuracy and sensitivity is improved for melanoma, leading to the detection of thinner tumors [9]. The diagnostic discrimination of actinic keratosis, Bowen’s disease/squamous cell carcinoma in situ, squamous cell carcinoma, and keratoacanthoma, which share common features, can be achieved using dermoscopy [12]. Further analysis suggests that dermoscopic images alone transferred via telemedicine are not satisfactory for remotely diagnosing skin cancer accurately [11]. For this reason, new methods are needed to improve skin cancer diagnosis that can be used with telemedicine to improve the ability to remotely diagnose skin lesions.

We have developed a new method termed vibrational optical coherence tomography (VOCT) to noninvasively characterize the type and margins of skin cancers [14,15,16,17,18,19]. VOCT provides a quantitative mechanovibrational spectrum that can be used to identify cellular components, blood vessels, dermal collagen, and fibrosis. New peaks associated with cancer associated cells, new friable blood vessels, and cancer associated fibrosis have also been identified [14,15,16,17,18,19,20,21]. Fingerprints of actinic keratosis (AK), basal cell carcinoma (BCC), squamous cell carcinoma (SCC), and melanoma have been identified and were shown to be significantly different from each other at a 0.95 confidence level [21].

Recent study results suggest that cancerous lesions are characterized by the addition of three new resonant frequency peaks [19,20,21]. A new cellular peak at 80 Hz with a modulus of 1.8 MPa, a new blood vessel peak at 130 Hz with a modulus of 4.10 MPa, and a new fibrous tissue peak at 260 Hz with a modulus of between 15–17 MPa are present in carcinomas. The 80 Hz and 130 Hz peaks are present in AKs but not in normal skin [14,15,16,17,18,19,20,21]. It is reported that the sensitivity and specificity of dermoscopy alone to detect melanoma varies from about 0.4 to 0.9 [22].

A new pathway for cellular migration may be laid down by cancer associated fibroblasts in the form of friable blood vessels and fibrous tissue [21]. Fibrous tissue appears to surround the new cells and new lesion blood vessels, based on OCT images and pixel intensity versus depth data [16,21]. Information on the metastatic potential of a lesion may be derived from the morphology and location of the fibrous tissues in relation to new cancer-associated cells and lesion blood vessels.

The purpose of this pilot study is to use machine learning to use the height and location of peaks in the mechanovibrational spectrum to differentiate between normal skin and different cancerous tissues. VOCT data can be collected remotely and, along with machine learning, can be used to noninvasively predict the specificity and sensitivity of differentiating between normal skin and different cancers. Ultimately, VOCT data, visual observation, dermoscopy, and machine learning and artificial intelligence can be used with telemedicine to noninvasively identify potentially malignant skin cancers in remote areas of the country where dermatologists are not readily available.

## 2. Materials and Methods

### 2.1. Subjects

Normal skin (N = 80) was studied in vivo using VOCT after informed consent was obtained, as reported previously [14,15,16,17,18,19,20,21]. The data were collected in the reflectance mode for both normal skin and for skin biopsies. Both experiments used a sinusoidal sound wave applied transversely to the skin. Control skin was examined from the hands, arms, and legs of each subject, as described previously [16,17,18,19,20,21]. The control subjects studied ranged in age from 21 to 71 years old. Tissue biopsies were studied from patients undergoing excisional therapies, as described previously [16,17,18,19,20,21].

### 2.2. Measurement of Resonant Frequency

Measurements were made using an OQ LabScope 2.0 that was modified by adding a 2 inch-diameter speaker placed about 2.5 inches from the area to be studied. An app provided a sinusoidal sound wave at 55 dB SPL that is part of the LabScope. The sound wave and the light were applied transverse to the surface of the sample along the axis of the light beam, as described previously [16,17,18,19,20,21].

Raw image data were collected using LabScope 2.0 that was used to calculate sample displacements from A line data. The data were processed using MATLAB software, as discussed previously [14,15,16,17,18,19,20,21].

The measured resonant frequencies were converted into elastic modulus values using a calibration Equation (1) [14,15,16,17,18,19,20,21]. The peak frequency (the resonant frequency), fn, is defined as the frequency at which the displacement is maximized.

A normalized weighted displacement value was generated, as discussed previously [16,17,18,19,20,21]. Sample component displacements are inversely related to the modulus) in MPa of the tissue elements, where fn is the resonant frequency and d is the sample thickness in m.
E × d = 0.0651 × (fn)^2^ + 233.16(1)

Normal skin studies were conducted in vivo, as described previously, as well as tissue biopsies in vitro [16,17,18,19,20,21]. Results of previous studies on skin lesions in vitro and in vivo indicated that the resonant frequency of tissue components were similar both in vitro and in vivo [17]. Histopathology was conducted by a dermatopathologist after routine processing [16,17,18,19,20,21]. Mohs thin sections were prepared by frozen sectioning and staining with H&E. The histopathological diagnosis was compared to the distribution of resonant frequency peaks, as described previously [21].

### 2.3. Machine Learning Analysis

Different machine learning algorithms, such as logistic regression, support vector, and decision making models, were implemented and compared to find the appropriate model yielding the highest accuracy. The logistic regression model showed the highest accuracy of the machine learning algorithms. Three distinct datasets were inputted into the machine. Each dataset consisted of weighted displacement for selected frequencies of 50 Hz, 80 Hz, 100 Hz, 130 Hz, and 260 Hz as features for the machine to learn. The datasets for different cancerous lesions (BCC, SCC, and melanoma) were compared to normal skin (controls). Each of the datasets contained 80 samples in total, split equally into 40 controls and 40 lesions. For each dataset, 80% of the samples were used to train the machine while 20% were used to test the prediction accuracy of the model. Using these measurements, the sensitivity and specificity were calculated based on Teventhan [23].

## 3. Results

Experiments were conducted on normal skin in vivo and excisions of BCC, SCC, and melanoma in vitro. New resonant frequency peaks in skin cancers were observed at 80 Hz, 130 Hz, and 260 Hz, as reported previously (see Figure 1, Figure 2, Figure 3 and Figure 4) [17,18,19,20,21]. This data was used in this study to carry out machine learning.

Figure 1 shows a typical color-coded cross-section of normal skin and a plot of weighted displacement versus frequency. Resonant frequency peaks are seen at 60 Hz, 100 Hz, and 260 Hz. Since the sound frequencies are incremented in 10 Hz intervals, the peak locations are only accurate to 10+/− Hz. Figure 2 shows a cross-sectional image and plot of weighted displacement versus frequency for a typical nodular BCC. A cross-sectional image and plot of weighted displacement versus frequency for SCC is shown in Figure 3, while Figure 4 shows a cross-sectional image and plot of weighted displacement versus frequency for a melanoma. Average peak heights for normal skin, basal cell carcinoma (BCC), squamous cell carcinoma (SCC), and melanoma for the 80, 130, and 260 peaks are shown in Figure 5. Note that only BCC, SCC, and melanoma have significant resonant frequency peaks at 80, 130, and 260 Hz.

The peak heights (see Figure 5) of new cells at 80 Hz, new blood vessels at 130 Hz, and fibrosis at 260 Hz seen in cancerous skin lesions are larger than those found in normal skin. Peaks at 80, 130, and 260 Hz, are “fingerprints” of BCC, SCC, and melanoma, indicating the presence of new cells at 80 Hz, new lesion blood vessels at 130 Hz, and fibrous tissue at 250–260 Hz, as reported previously [16,17,18,19,20,21].

Normalized peak heights (50, 80, 100, 130, and 260 Hz) of normal skin, BCC, SCC, and melanoma were compared using machine learning to determine whether differences in the peak heights and locations that were used to train the computer could be used to recognize differences between cancerous lesions and normal skin. Table 1 shows the results of the analysis of the location and heights of the vibrational spectrum of the different cancerous lesions. Preliminary results listed in Table 1 illustrate that the specificity and sensitivity of predicting the differences between normal skin and the cancerous lesions SCC and melanoma were 92% and 83% and 88% and 78%, respectively (see Table 1). The values of the sensitivity and specificity of predicting the difference between normal skin and BCC were 91% and 88%.

The first dataset consisted of 40 samples of both BCC and normal skin, of which 61 samples were used to train the model and 19 samples were used to test the models. Machine learning was accurately able to predict 7 out of 7 of the BCC samples, and 10 out of 12 of the normal skin samples. Hence, the machine had an average accuracy of 89.47%.

Similarly, for the SCC vs. the normal skin dataset, 60 samples were used to train the machine while 20 were used to test the model. Machine learning was accurately able to predict 11 out of 12 SCC samples and 7 out of 8 normal skin samples. Hence, the machine had an average accuracy of about 90%. For the dataset consisting of melanoma data, 60 samples were used to train the machine, whereas 20 were used to test the model. Machine learning was accurately able to predict 9 out of 9 of the melanoma samples and 10 out of 11 of the normal skin samples correctly. This resulted in the average accuracy of the model being about 95%.

## 4. Discussion

Skin lesion analysis is performed by specialized dermatologists through visual and dermoscopic evaluations. However, these techniques have a poor diagnostic ability and over 2.5 million benign skin biopsies (55% of the total) are performed each year in the US at an alarming cost of USD ~2.5 B [24,25]. New methods are needed to noninvasively differentiate lesions that need to be biopsied from benign lesions.

The results of previous studies have shown that cancerous lesions can be differentiated from normal skin based on the addition of new cells with increased stiffness (80 Hz), new lesion blood vessels that appear to be less stiff (130 Hz), and fibrous tissue (250–260 Hz) that is very stiff that are present in all carcinomas [21].

VOCT data can be remotely collected over the internet after the area of skin to be studied is located visually. The only requirement for the “on site” data collection is to focus the visual camera that is part of the VOCT handpiece on the skin. This can be performed by a worker on site after training for less than 1 h. The remainder of the image and VOCT data collection can be performed by a technician located at the collection and data processing “home” location. Software loaded onto the VOCT computer can run the instrument and download the collected data for analysis by a trained physician at the “home” site.

It is important to be able to noninvasively predict the invasive potential of skin cancers, including SCC and melanoma, based on the early evaluation of lesions that may be difficult to diagnose. Using VOCT, it is possible to capture OCT images and quantitative resonant frequency information on lesions of less than 0.6 mm. Using VOCT images and quantitative resonant frequency observations, lesions as small as 0.10 mm can be identified. These lesions cannot be clearly observed by visual inspection and dermoscopy. In this pilot study we used machine learning to attempt to differentiate between normal skin, BCC, SCC, and melanoma. This differentiation is needed to be able to noninvasively characterize skin lesions and to determine the ability to use VOCT data to positively identify a malignant lesion before it reaches 0.6 mm, a size thought to be critical to limit invasiveness.

The results suggest that machine learning using VOCT is a potential method for screening skin lesions for invasive potential, noninvasively, without the bias of human analysis. This study was conducted by comparing training the computer using VOCT data derived from peak heights at various sound frequencies to differentiate skin cancers from normal skin. While this approach shows merit, we have recently demonstrated that the use of peak height ratios of 50 Hz/80 and 130 Hz/80 Hz may give better results in defining the lesion type [21]. Further studies are underway to evaluate the use of peak height ratios to better define differences between normal skin and cancerous lesions.

An improved margin and depth analysis can be obtained using VOCT using a higher power laser for very large deep lesions. In addition, the use of ultrasound in conjunction with VOCT has been demonstrated to characterize deeper tissues, including tendon, muscle, blood vessels, bone, and nerve [20]. The sound will penetrate up to about 8 CM so that deep lesions can be identified using ultrasound in conjunction with VOCT.

The results of this pilot study are encouraging and should be able to be improved after additional lesion data is collected. Providing a larger data pool would likely improve the comparison between normal skin and skin cancers, as well as considering the use of peak ratios, especially 50/80 Hz and 130/80 Hz. Ongoing studies are underway to increase the database, as well as use of other AI methods to discriminate between normal skin and skin cancers.

The results of this pilot study show that the use of machine learning and the height of the mechanovibrational peaks collected using VOCT has potential for being able to noninvasively differentiate between normal skin and different cancerous lesions. Noninvasive VOCT data, along with machine learning, is shown to predict the specificity and sensitivity of VOCT for differentiating between normal skin and different skin cancers at rates between 78% and 92%, which may be improved using larger data sets and other AI techniques. Ultimately, VOCT data, visual inspection, and dermoscopy, in conjunction with machine learning, may be used with telemedicine to noninvasively identify small potentially malignant skin cancers in remote areas of the country where dermatologists are not readily available.

## 5. Conclusions

We have used machine learning and the height and location of the VOCT mechanovibrational peaks collected on skin biopsies to noninvasively differentiate between normal skin and different cancerous lesions. Vibrational optical coherence tomography data, along with machine learning, is shown to noninvasively predict the specificity and sensitivity of differentiating between normal skin and different skin cancers at rates between 78 and 92%. The sensitivity and specificity may be improved using larger databases and other peak ratios. Ultimately, VOCT data, visual inspection, and dermoscopy, in conjunction with AI, may be used in telemedicine to identify small potentially malignant skin cancers noninvasively in remote areas of the country where dermatologists are not readily available.

## Figures and Tables

**Figure 1 cancers-15-00156-f001:**
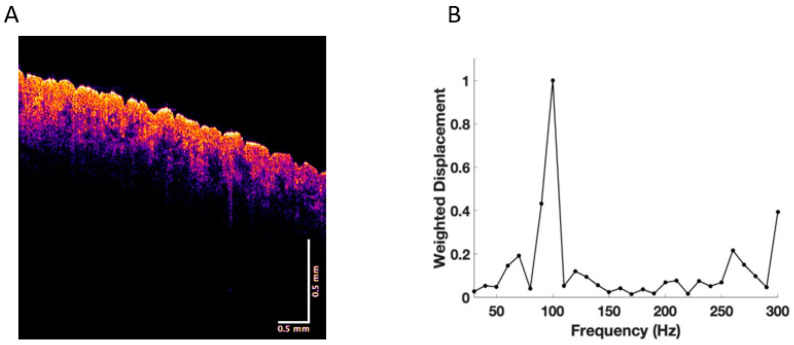
(**A**) Color-coded OCT image of normal skin showing the epidermis in yellow and pink and the papillary dermis in blue. (**B**) Plot of weighted displacement versus frequency for normal skin showing the cellular (60–80 Hz), papillary collagen fibers (100), and a small fibrotic peak (250–260 Hz).

**Figure 2 cancers-15-00156-f002:**
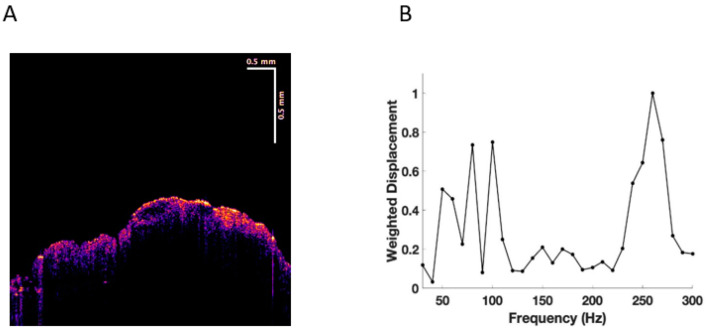
(**A**) Color-coded OCT image of a nodular basal cell carcinoma (BCC) showing the thin epidermis (yellow and pink) and the papillary dermis in blue. Note: the black oval spots are the location of the lesions. (**B**) Plot of weighted displacement versus frequency for BCC showing the new cellular (80 Hz), papillary collagen fibers (100–120 Hz), and new large fibrotic peak (250–260 Hz).

**Figure 3 cancers-15-00156-f003:**
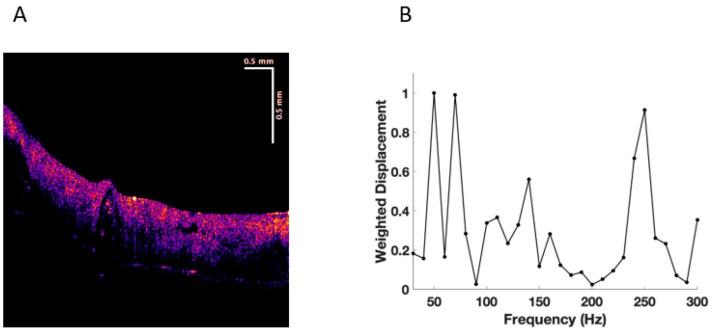
(**A**) Color-coded OCT image of a squamous cell carcinoma (SCC) showing the paucity of epidermis (yellow and pink) and the papillary dermis in blue. Note: the black spots are the location of the lesions. (**B**) Plot of weighted displacement versus frequency for SCC showing the new cellular (80 Hz), new vascular peak (130 Hz), and large fibrotic peak (250–260 Hz).

**Figure 4 cancers-15-00156-f004:**
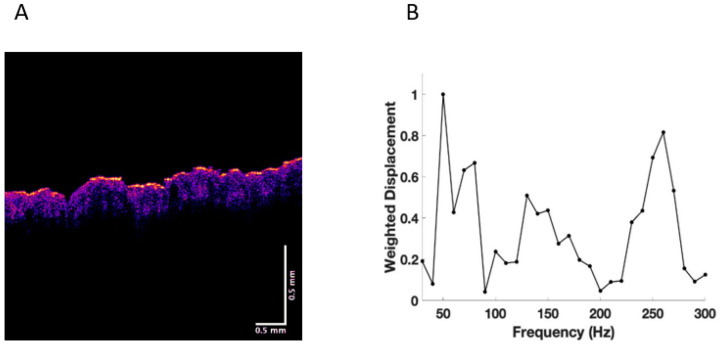
(**A**) Color-coded OCT image of a melanoma showing the paucity of epidermis (yellow and pink) and the papillary dermis in blue. Note: black spots show the location of the lesions. (**B**) Plot of weighted displacement versus frequency for SCC showing the new cellular (80 Hz), new vascular peak (130 Hz), and large fibrotic peak (250–260 Hz).

**Figure 5 cancers-15-00156-f005:**
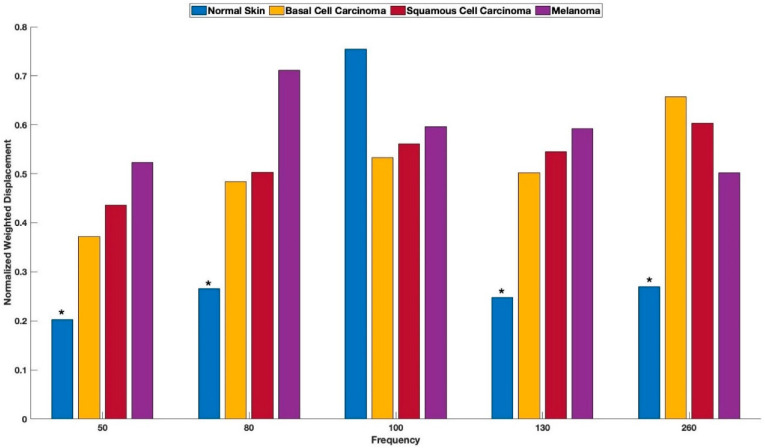
Normalize average peak heights for normal skin, BCC, SCC, and melanoma. Note: the asterisks denote statistical difference at a 0.95 level using a paired two tailed student’s *t* test of normal skin and all the cancerous lesions.

**Table 1 cancers-15-00156-t001:** The sensitivity and the specificity of the three different cancerous tissues trained using 60 samples, when compared with normal skin. The sensitivity and specificity equations used were defined previously [23].

	BCC	SCC	Melanoma
Sensitivity	90.9%	91.6%	83.3%
Specificity	87.5%	87.5%	77.8%

## Data Availability

Data contained in this study can be found at optovibronex.com, accessed on 15 July 2022.

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
