# Peer review of "Identification of Cancerous Skin Lesions Using Vibrational Optical Coherence Tomography (VOCT): Use of VOCT in Conjunction with Machine Learning to Diagnose Skin Cancer Remotely Using Telemedicine"

_cancers, 2022, doi:10.3390/cancers15010156_

Round 1

Reviewer 1 Report

This is an approach to diagnose skin cancers using non invasive tool. This can be proved useful in the case diagnostic accuracy is good enough. As a clinician involved in the field of skin cancers and imaging of the skin, I have to say that dermoscopy offers better accuracy than VOCT. The authors should state this in the limitation section. However, I do agree that artificial intelligence combining dermoscopy with another technique which is able to provide some insight within deeper layers  of the skin might be more accurate, and finally useful in real life.

Another limitation is that skin cancers are easily removed allowing accurate pathology diagnosis and treatment.

Fig. 2. The legend indicates “the new cellular (80 Hz), papillary collagen fibers (100 Hz), and new large fibrotic peak (260 Hz). “What is the level of evidence that these peaks are really related to collagen fibers or fibrosis?  I suggest to the authors they add  clinical picture or at least pathology slide of the lesion.

Determining margins of the lesion is another crucial point especially in the case the lesion is not surgically removed. From the pictures, I am not convinced that lateral and deep margins can be determined using the technique. Can the author discuss this point?

Determining the thickness of tumour, especially for melanomas,  is also a key issue. Was this possible with the technique used? Can the author discuss this point? Is there another non-invasive tool, e.g. ultrasound, allowing this measurement, allowing one-step surgery?

Author Response

We thank the reviewer for their helpful comments and believe the revised manuscript is much improved and is ready for publication. The responses are given below in red.

Fred Silver

Review #1

This is an approach to diagnose skin cancers using non invasive tool. This can be proved useful in the case diagnostic accuracy is good enough. As a clinician involved in the field of skin cancers and imaging of the skin, I have to say that dermoscopy offers better accuracy than VOCT. The authors should state this in the limitation section. However, I do agree that artificial intelligence combining dermoscopy with another technique which is able to provide some insight within deeper layers of the skin might be more accurate, and finally useful in real life.

Thank you for pointing out that we need to add to the text a reference quantifying detection of melanoma using dermoscopy. According to Dinnes et al., 2018, the specificity and sensitivity of melanoma detection ranges from 0.4 to 0.9. This information has been added to the introduction. Clearly even with only a limited number of specimens used in this pilot study, detection with VOCT appears to equal or exceed those numbers.

Another limitation is that skin cancers are easily removed allowing accurate pathology diagnosis and treatment. Unfortunately, many patients do not want a skin excision on their face, head, or neck if it is not necessary. In the US 55% of all skin biopsies are benign so limiting the number of skin excisions for patients unwilling to undergo skin excision of benign lesions is desirable. This information has been added to the revised manuscript.

Fig. 2. The legend indicates “the new cellular (80 Hz), papillary collagen fibers (100 Hz), and new large fibrotic peak (260 Hz). “What is the level of evidence that these peaks are really related to collagen fibers or fibrosis?  I suggest to the authors they add clinical picture or at least pathology slide of the lesion.

We have published several papers describing the relationship between the new resonant frequencies seen in the lesions see references 15-22. In addition, we have grown cells in culture, isolated dermal collagen and studied fibrosis in wounds during wound healing to define these peaks.

Determining margins of the lesion is another crucial point especially in the case the lesion is not surgically removed. From the pictures, I am not convinced that lateral and deep margins can be determined using the technique. Can the author discuss this point?

We have added a discussion to the revised manuscript concerning lateral and depth margins. Deep margins can be determined using a higher power laser for very large lesions. In our study we studied very small lesions and used a lower power light source.

Determining the thickness of tumour, especially for melanomas, is also a key issue. Was this possible with the technique used? Can the author discuss this point? Is there another non-invasive tool, e.g. ultrasound, allowing this measurement, allowing one-step surgery?

Thank you, that is an important point. We have used ultrasound in conjunction with VOCT to study  deeper tissues including tendon, muscle, blood vessels, bone and nerve. The sound will penetrate up to 8 CM while the infrared light will only penetrate tissue to 1 to 2 mm. We have added a reference to discuss measuring lesion depth.

Reviewer 2 Report

Although the data-gathering method has been described clearly, the ML method needs more description and clarification.

Author Response

Review #2

Although the data-gathering method has been described clearly, the ML method needs more description and clarification.

Thank you for this comment. We have added additional details on the ML method to the manuscript.

Round 2

Reviewer 2 Report

Thank you for applying the previous comments, still it needs scientific and language edits.